# AsiaRiceYield4km: Seasonal Rice Yield in Asia from 1995 to 2015

Huaqing Wu[1,2]★, Jing Zhang[1,3]★, Zhao Zhang[1,2], Jichong Han[1,2], Juan Cao[1,2], Liangliang Zhang[1,2], Yuchuan Luo[1,2], Qinghang Mei[1,2], Jialu Xu[2], Fulu Tao[4,5]

[1]Key Laboratory of Environmental Change and Natural Disasters, Ministry of Education Beijing Normal University, Beijing 100875, People's Republic of China
[2]School of National Safety and Emergency Management, Beijing Normal University, Beijing 100875 / Zhuhai 519087, People's Republic of China
[3]Faculty of Geographical Science, Beijing Normal University, Beijing 100875, China
[4]Key Laboratory of Land Surface Pattern and Simulation, Institute of Geographical Sciences and Natural Resources Research, Chinese Academy of Sciences, Beijing, 100101, People's Republic of China
[5]College of Resources and Environment, University of Chinese Academy of Sciences, Beijing 100049, People's Republic of China

★These authors contributed to the work equally and should be regarded as co-first authors.

*Correspondence to*: Zhao Zhang (zhangzhao@bnu.edu.cn)

**Abstract.** Rice is the most important staple food in Asia. However, high-spatiotemporal-resolution rice yield datasets are limited over this large region. The lack of such products greatly hinders studies that are aimed at accurately assessing the impacts of climate change and simulating agricultural production. Based on annual rice maps in Asia, we incorporated multi-sources predictors into three machine learning (ML) models to generate a high-spatial-resolution (4km) seasonal rice yield dataset (AsiaRiceYield4km) from 1995 to 2015. Predictors were divided into four categories that considered the most comprehensive rice growth conditions and the optimal ML models was determined based on an inverse proportional weight method. The results showed that AsiaRiceYield4km achieves good accuracy for seasonal rice yield estimation (single rice: $R^2$ = 0.88, $RMSE$ = 920 kg/ha, double rice: $R^2$ = 0.91, $RMSE$ = 554 kg/ha, and triple rice: $R^2$ = 0.93, $RMSE$ = 588 kg/ha). Compared with single rice of Spatial Production Allocation Model (SPAM), the $R^2$ of AsiaRiceYield4km was improved by 0.20 and $RMSE$ was reduced by 618 kg/ha on average. In particular, constant environmental conditions including longitude, latitude, elevation, and soil properties contributed the most (~45%) to rice yield estimation. For different rice growth periods, we found that the predictors of the reproductive period had greater impacts on rice yield prediction than those of the vegetative period and the whole growing period. AsiaRiceYield4km is a novel long-term gridded rice yield dataset that can fill the unavailability of high-spatial-resolution seasonal yield products across major rice production areas and promote more relevant studies on agricultural sustainability worldwide. AsiaRiceYield4km can be downloaded from an open-data repository (DOI: https://doi.org/10.5281/zenodo.6901968; Wu et al., 2022).

**1 Introduction**

As one major staple crop, rice (*Oryza sativa* L.) provides more than a quarter of calories for approximately half of the population with only 11% of the arable land on the earth (Maclean et al., 2002; Alexandratos and Bruinsma, 2012; Birla et al., 2017; Qian et al., 2020). Asia produces and consumes more than 90% of the global rice (Bandumula, 2018), which is dominated by poor smallholder farmers. Therefore, information on rice yield in Asia is essential for sustaining food security and farmers' livelihoods (Laborte et al., 2017). In the last half-century, the growth of rice yields has contributed more to an increase in production than the expansion of planting areas (Blomqvist et al., 2020) and will remain a dominant factor considering the land-use policies for reducing environmental pressure (Lambin and

Meyfroidt, 2011; Kim et al., 2021). In addition, Asia has complex rice cropping systems where rice may

be cultivated multiple times within one year (Zhang et al., 2020a). It is critically necessary to identify the

long-term and seasonal Asia rice yields – at high spatial resolution to monitor and guide agricultural

production.

Previous global-scale crop yield datasets, including Harvester Area and Yields of 175 crops

(M3Crops) (Monfreda et al., 2008), Spatial Production Allocation Model (SPAM) (You and Wood, 2006;

Yu et al., 2020), Global Dataset of Historical Yields of Major Crops (GDHY) (Iizumi et al., 2014; Iizumi

and Sakai, 2020), and Global Gridded Crop Model Intercomparison (GGCMI) phase 1 (Müller et al.,

2019), have been produced and widely employed in many studies (Folberth et al., 2020; Kaltenegger and

Winiwarter, 2020; Iizumi et al., 2021; Lin et al., 2021; Liu et al., 2021b). However, due to the different

research goals and technical restrictions, their spatial resolutions are relatively coarser (e.g. ~10km for

M3Crops and SPAM; ~55km for GDHY and GGCMI phase 1) and temporal resolutions are mostly

annual (Laborte et al., 2017). Only a few datasets have seasonally temporal information (e.g., GDHY)

but still cannot cover all rice seasons (Kim et al., 2021). In addition, the time spans are limited (e.g.,

only one year for M3Crops; every five years for SPAM). For the long-term rice yield dataset, GDHY,

the authors used a fixed rice area basemap that did not obtain the interannual spatial dynamics of rice

yield. To the best of our knowledge, a long-term seasonal rice yield dataset with higher spatial resolution

and dynamic spatial distribution is currently unavailable for the major rice planting regions on the world.

To address the above issues, there is a significant need to acquire  multi-sources data and wiser

technologies for rice yield estimation (Chlingaryan et al., 2018; Cao et al., 2020; van Klompenburg et

al., 2020; Zhang et al., 2020b; Chen et al., 2022). With the rapid development of remote sensing

technology in recent years, large-scale and long-term high-spatiotemporal observations provide ample

and timely phenological and growing information for rice growth. Ground-based data such as climate

and soil also provide more key environmental information (Folberth et al., 2016; Zhang et al., 2021).

Many publications that successfully combine satellite-derived data and ground environmental

information for yield estimation have expanded our knowledge (Huang et al., 2013; Mosleh et al.,

2015; Cao et al., 2021; Fernandez-Beltran et al., 2021). Nevertheless, few studies have yet employed

annual paddy rice areas for yield estimation. Moreover, machine learning (ML), such as random forest

(RF), extreme gradient boosting (XGBoost), and long short-term memory (LSTM) has been

increasingly and successfully used in crop yield estimation (Cai et al., 2019; van Klompenburg et al., 2020; Sakamoto, 2020; Luo et al., 2022). Such ML models can overcome the drawbacks of two traditional estimation methods: process-based crop models (PCMs) and statistical regression methods (SRMs). Compared with PCMs, ML can wisely select input variables according to the actual requirements and local geographical environment conditions without complicated parameters (Jeong et al., 2022). Due to the complex functions with higher efficiency and flexibility, the yield estimation results of ML are always better than those of SRMs (Chlingaryan et al., 2018). In addition, ML has a good spatial generalization. Therefore, ML models combined with multi-sources data potentially provide a good chance for large-scale gridded yield estimation and their accuracy improvement.

Overall, we would integrate multi-source data and annual rice maps into ML models for generating a seasonal rice yield dataset at 4km resolution across Asia (AsiaRiceYield4km) from 1995 to 2015. AsiaRiceYield4km will better support agricultural monitoring systems and related research over a large scale because of its higher-spatiotemporal resolution and longer-time span.

## 2 Materials and methods

### 2.1 Study area

Asia is the most important rice-producing area accounting for 89% of the planting area and 91% of the global production (Food and Agriculture Organization of the United Nations, FAO, 2022). Considering the accessibility of locally census-based rice yield data, 14 main rice-producing countries of Asia were selected and then divided into 27 cases (one case refers to one specific rice-cropping period in a country) based on different rice cropping systems (single, double, and triple rice), as shown in Fig. 1.

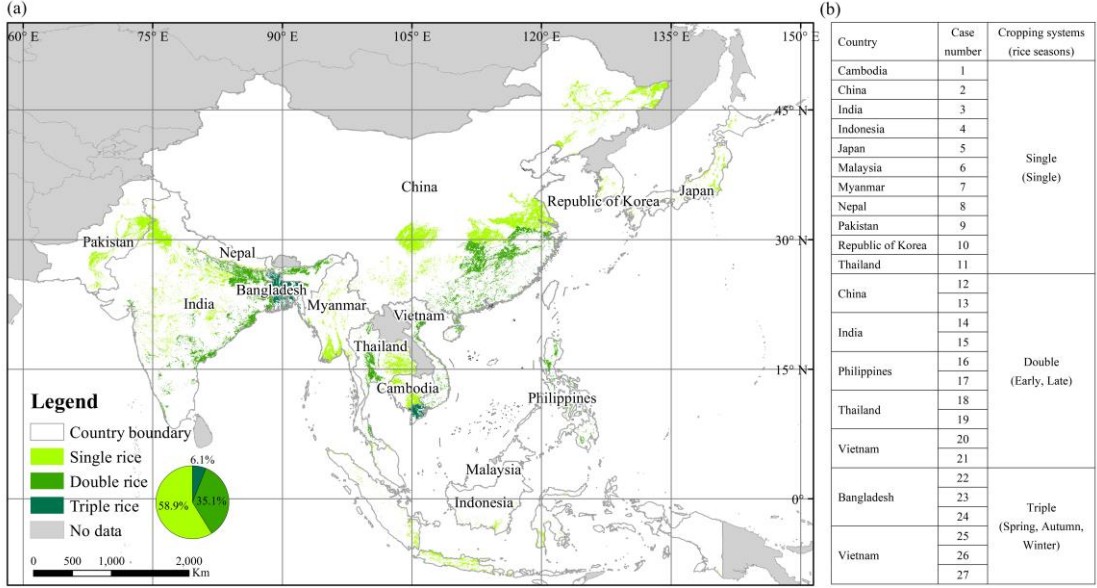

**Figure 1: (a) Rice planting areas with different cropping systems in the main rice-producing countries of Asia. The green area represents the maximum paddy rice area where paddy rice grew for at least one year during the period 1995-2015 (Han et al., 2021, 2022). The pie chart represents the area proportion of different rice cropping systems. (b) Case numbers and cropping system for each country. Double rice follows the order of early before late (e.g., 12 and 13 represent the early season rice and late season rice in China, respectively), and triple rice follows the order of spring, autumn, and winter (e.g., 25, 26, and 27 represent the spring season rice, autumn season rice and winter season rice of Vietnam, respectively).**

## 2.2 Data

Multi-sources data were collected for rice yield estimation, including annual rice area maps, rice yield of 1400 administrative units (minimum administrative division scale units for each country with available rice yield), leaf area index (LAI) information from remote sensing products, and rice growth environmental conditions (location, time, soil, and climate). In addition, considering the necessity of phenological information, we also produced gridded key phenological dates from LAI data based on inflection-based and threshold-based methods (Sect. 2.3.1). Except for yield records at administrative unit scale from official statistics (Table S1), the other data were resampled to 4km×4km by the nearest neighbour resampling method in ArcMap 10.2 (originally spatial information is listed in Table S2).

### 2.2.1 Rice area maps

We selected the latest public rice distribution map dataset, APRA500 (annual dataset of paddy rice area at a 500m resolution from 2000 to 2020), in this study (Han et al., 2021, 2022). APRA500 has annual rice distribution information which can reduce the influence of other land cover types. Due to the

topography conditions, cloud contamination, and the mixed-pixel effects with fragmented cropland fields,

rice area in APRA500 was somehow underestimated (Han et al., 2022). To reduce this effect, we used

the rice area union of the three years (current year, last year, and next year) to represent the rice area of

the current year (e.g., the area of 2005 is the union of 2004, 2005, and 2006). Specifically, the union area

of 2000, 2001, and 2002 was also applied to the years from 1995 to 2000 because of the unavailable area

maps.

**2.2.2 Seasonal rice yield**

Rice seasons were determined mainly based on RiceAtlas (Laborte et al., 2017). RiceAtlas is the most

comprehensive and detailed database for rice season and has been widely used in many studies (van Oort

and Zwart, 2018; Muehe et al., 2019; Fritz et al., 2019). The United States Department of Agriculture

(USDA, https://ipad.fas.usda.gov/ogamaps/cropcalendar.aspx, last accessed: 7 April 2022) and the

national statistics of each country were also referenced for rice seasons determination. The rice seasons

have various names in different countries, such as Aman, Aus, and Boro for triple rice of Bangladesh

and Rabi and Kharif for double rice of India. To make the data more readable and consistent, we used

single rice (single season), double rice (early and late seasons), and triple rice (spring, autumn, and winter

seasons) for the three rice cropping systems in our study, as shown in Fig.1b. A few rice seasons (e.g.,

the early season in Cambodia, Malaysia, Myanmar, and Indonesia; and the winter season in India) were

not considered due to the lack of yield records.

We collected seasonal rice yield data from FAO and other government websites (Table S1). Over

45000 rice yield records of 1400 administrative units from 1995 to 2015 were collected. The quality of

these data has been checked and some yield outliers were filtered out according to the following rules:

(a) exceeding the actual biophysically attainable yields and (b) beyond the averages ± two times variance

during the period 1995-2015 (Zhang et al., 2014; Cao et al., 2020, 2021).

**2.2.3 Key phenological dates**

The transplanting, heading, and maturity dates are the three most important phenological dates

during rice growing period. The whole growing period (WGP) is divided into two periods according to

the three key phenological dates: vegetative period (VEP, from transplanting to heading) and

reproductive period (REP, from heading to maturity). However, most rice phenology datasets are always

at administrative scales without interannual variation. The USDA provided country-scale growing phenological information. RiceAtlas had subnational phenology information but disregarded the annual dynamics (Laborte et al., 2017). In addition, these datasets lack heading date information about rice. Here, we retrieved the three dynamic key rice phenological dates from remote sensing data in Asia during the period 1995-2015 at a 4km×4km grid scale by inflection-based and threshold-based methods (Sect. 2.3.1). The USDA and RiceAtlas datasets provided a threshold range for phenology and were used to validate our extracted phenological dates.

### 2.2.4 Location and time

Location information includes longitude (*Lon*), latitude (*Lat*), and elevation (*Ele*). The Global 30-arc-second (1km) gridded Digital Elevation Model (DEM) dataset (1999) from the National Oceanic and Atmospheric Administration (NOAA) was employed in this study. The *Lon* and *Lat* information was collected from the centroid of each resampled 4km pixel by ArcMap 10.2. The temporal information is represented by the year (1995-2015).

### 2.2.5 Soil data

Soil properties are important factors controlling rice growth and final yield. The Harmonized World Soil Database (HWSD) v1.2 provides key soil property variables, including: Topsoil Sand Fraction (*T_Sand*), Topsoil Silt Fraction (*T_SILT*), Topsoil Clay Fraction (*T_CLAY*), Topsoil Reference Bulk Density, (*T_BULK_DEN*), Topsoil Organic Carbon (*T_OC*), and Topsoil pH (H2O) (*T_PH_H2O*) (https://www.fao.org/soils-portal/soil-survey/soil-maps-and-databases/harmonized-world-soil-database-v12/en/, last accessed: 7 April 2022; Wieder et al., 2014).

### 2.2.6 Climate data

TerraClimate (Abatzoglou et al., 2018), a monthly high spatial resolution (4km) meteorological dataset (http://doi.org/10.7923/G43J3B0R, last accessed: 7 April 2022) from 1995 to 2015, was used in our study. This dataset provides climate and water balance information for Asia rice (Salvacion, 2022), including Palmer Drought Severity Index (*PDSI*), precipitation accumulated (*Pre*), downward surface shortwave radiation (*Srad*), maximum temperature (*Tmax*), minimum temperature (*Tmin*), vapor pressure (*Vap*), and wind speed (*Ws*).

**2.2.7 LAI**

Remote sensing indices have been widely used in rice yield estimation (Son et al., 2020; Arumugam et

al., 2021), but few studies have been conducted before 2000 (Liu et al., 2021a). To extend the period of

the gridded yield dataset from 1995 in this study, we adopted Global Land Surface Satellite (GLASS)

Advanced Very-High-Resolution Radiometer (AVHRR) LAI data (http://glass.umd.edu/Download.html,

last accessed: 7 April 2022; Xiao et al., 2013, 2016, 2017), which begun from 1981 with a fine spatial

resolution of 4 km and temporal resolution of 8 days. Compared with other similar products, GLASS

AVHRR LAI has the highest accuracy and lowest uncertainty (Liang et al., 2021). The GLASS AVHRR

LAI was used for rice phenological information extraction and yield estimation.

**2.3 Methods**

We applied three steps to generate AsiaRiceYield4km by incorporating multi-sources data into three ML

methods: determining phenological dates, categorizing and selecting predictors, and developing the

optimal models and generating gridded rice yield (Fig. 2). Details of each step are provided in the

following sections.

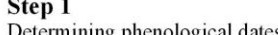

**Step 1**
Determining phenological dates

Figure flowchart showing Step 1 (Determining phenological dates), Step 2 (Categorizing and selecting predictors), and Step 3 (Developing the optimal models and generating gridded rice yield).

**Figure 2: Flowchart for generating long-term and high-resolution gridded rice yields by incorporating multi-sources data into ML models for one case. All 27 cases followed these steps and were combined to get the AsiaRiceYield4km dataset.**

### 2.3.1 Determining phenological dates

The inflection-based method (Chen et al., 2016; Luo et al., 2020a) and threshold-based method (Manfron et al., 2017) were employed to detect rice phenological dates (Fig.2 step1) according to the following rules: (1) Transplanting dates: The LAI always maintains a low value for a period before the transplanting date and dramatically increases after this date (Sakamoto et al., 2005; Chen et al., 2018). Therefore, if there is one point in the LAI curve where the first derivative is > 0 after it or its second derivative is equal

to 0, this point is defined as the transplanting date. (2) Heading dates: the inflection point from VEP to REP (Wang et al., 2018) is characterized by the maximum value of the LAI between the transplanting date and the maturity date (Son et al., 2013). (3) Maturity dates: the physiological activity of rice will sharply drop during the harvesting period. The first inflection point at the LAI curve where its first derivative becomes negative is considered the maturity date. In addition, LAI values of pixels beyond the averages ± two times standard deviation (SD) were filtered (Zhang et al., 2022). If the phenological dates in some grids cannot be detected by the above rules or be filtered, the average value of the administrative unit where the grids are located is applied.

### 2.3.2 Categorizing and selecting predictors

To provide comprehensive rice growth information for the ML models, we divided the multi-sources data into four categories including 50 predictors (Table S3): cumulative growing predictors of different growing periods (CGP), extreme growing predictors (EGP), constant environmental conditions (CEC), and temporal information (TI) (Fig. 2 step2). The CGP includes the sum of each LAI and climate variable in different growing periods (VEP, REP, and WGP), reflecting the overall growing and weather difference of the three continuous growing periods. The EGP consists of the maximum and minimum of each climate and LAI variable considering the impact of extreme events. CEC reflects the influence of the geographical environment on rice growth. TI reflects long-term agronomic technology improvements and variety renewal (Huntington et al., 2020). All these predictors were aggregated to administrative scale. The predictor values of grids located in one administrative unit were averaged to this administrative unit.

High-dimensional predictors often affect the accuracy and computational efficiency of ML methods (LeCun et al., 2015; Zhang et al., 2019). To reduce this effect, Pearson correlation analysis was employed to estimate the relationship between yield and other variables for each case. The variables with a significant correlation ($p < 0.05$) were selected as predictors (Cao et al., 2021). The yield and selected predictors of one case were input into one model. Specifically, the four predictors, *Lon*, *Lat*, *Ele,* and *Year*, were considered to have a stable impact on rice yield and were included in all 27 estimation models for the 27 cases (Ray et al., 2019; Huntington et al., 2020). Considering the covariate-relation of the

predictors in CGP for WGP and the remaining two periods, the predictors of WGP would be selected if its Pearson *R* was higher than that in the remaining two periods, or vice versa.

**2.3.3 Developing the optimal models and generating gridded rice yield**

(1) Dataset division rules

To effectively reduce overfitting effects (Dinh and Aires, 2022), we divided all data into three sets

(training, validation, and testing) which were used to optimize the ML parameters, select the optimal model, and evaluate its generalization ability, respectively (Ripley, 2007). The diagram of the database division process is shown in Fig. 2 step3. For each case, the whole database contained the selected predictors from all administrative-scale units during 1995-2010. The database was randomly divided into two subsets by the administrative unit: 20% of the samples were used for testing and the remaining 80%

were randomly resplit into 70% for training and 30% for validation without consideration of administrative units. Thus, the training, validation, and testing sets contain 56% (80%×70%), 24% (80% ×30%), and 20% (20%×100%), respectively, of the whole dataset. Such division rules avoid information leakage from the testing set to the training set (Meroni et al., 2021) and enhance the robustness of the model.

(2) ML models

ML can develop transfer functions based on the relationships between predictors and target variables for rice yield estimation (Chlingaryan et al., 2018; Shahhosseini et al., 2020). Three widely employed ML models, RF, XGBoost, and LSTM were selected for rice yield estimation. The RF is based on the bagging ensemble model, which generates multiple decision trees and obtains predictions by voting on all

individual trees (Breiman, 1996, 2001). In addition, extra randomness is introduced to the RF when generating trees and searching for the best tree stages (Shahhosseini et al., 2020). It provides more diversity for trees and can generate the overall better performance model (Zhang et al., 2019). XGBoost uses the optimized gradient boost for decision trees, which tries to make weak learners strong (Chen and Guestrin, 2016). This method adopts an updated strategy to train the estimated model and the updated

model minimizes the loss by reducing errors from previous models (Obsie et al., 2020). LSTM is a special recurrent neural network (RNN) that is proposed to overcome the vanishing and exploding gradient problems of RNNs (Hochreiter and Schmidhuber, 1997; Sak et al., 2014; Tian et al., 2021). LSTM

contains input, hidden and output layers and the hidden layers consist of memory cells (He et al., 2019; Zhang et al., 2019). Tuning hyper-parameters can effectively improve the accuracy for rice yield estimation (Shahhosseini et al., 2021). The hyper-parameters tuning details and Python library information of the ML algorithm are shown in Supplementary Methods.

(3) Model evaluation

The coefficient of determination ($R^2$) and root-mean-square error ($RMSE$) were adopted to evaluate the performance of each model for each case.

$$R^2 = 1 - \sum_{i=1}^{n} \left( Y_{i,j}^{ob} - Y_{i,j}^{es} \right)^2 / \sum_{i=1}^{n} \left( Y_{i,j}^{ob} - \bar{Y}_{i,j}^{ob} \right)^2 \tag{1}$$

$$RMSE = \sqrt{\sum_{i=1}^{n} \left( Y_{i,j}^{es} - Y_{i,j}^{ob} \right)^2 / n} \tag{2}$$

where $i$ is the number of administrative units, $n$ is the total number of administrative units, and $j$ is the year. $Y^{ob}_{i,j}$ is the observed rice yield from government or FAO websites in the $i$th administrative unit of year $j$, $\bar{Y}^{ob}_{i,j}$ is the average of the observed rice yield in the $i$th administrative unit of year $j$, and $Y^{es}_{i,j}$ is the AsiaRiceYield4km yield in the $i$th administrative unit of year $j$.

(4) The optimal yield estimation model selection

In this study, three ML models can generate three different yield estimation results. Previous studies recommend the weighted ensemble method by combining the estimation results of different methods, wishing for a relatively stable result but still giving up some accuracy (Shahhosseini et al., 2020, 2021). Moreover, many studies also selected the optimal ML model by comparing only the accuracy of validation/testing sets (Zhang et al., 2021; Chen et al., 2022; Luo et al., 2022). Here, to conduct a comprehensive evaluation of different ML models and datasets, we developed an inverse proportional weight (IPW) method to assign weights for training, validation, and testing accuracy to calculate the adjusted accuracy for each ML model (Eq. 3-7). The ML model with the best adjusted accuracy was selected as the optimal ML model.

$$w_{tr} = p_{tr} / \left( p_{tr} + p_{va} + p_{te} \right) \tag{3}$$

$$w_{va} = p_{va} / \left( p_{tr} + p_{va} + p_{te} \right) \tag{4}$$

$$w_{te} = p_{tr} / \left( p_{tr} + p_{va} + p_{te} \right) \tag{5}$$

$$R_{ad}^2 = R_{tr}^2 \cdot w_{tr} + R_{va}^2 \cdot w_{va} + R_{te}^2 \cdot w_{te} \tag{6}$$

$$RMSE_{ad} = RMSE_{tr} \cdot w_{tr} + RMSE_{va} \cdot w_{va} + RMSE_{te} \cdot w_{te} \tag{7}$$

where $tr$, $va$, and $te$ are abbreviations for training, validation, and testing; $p_{tr}$, $p_{va}$, and $p_{te}$ are the inverse

proportions for the sizes of the training, validation, and testing sets, respectively, which are equal to

$1/0.56$, $1/0.24$, and $1/0.20$, respectively; and $w_{tr}$, $w_{va}$, and $w_{te}$ are the weights of the training, validation,

and testing sets, respectively. $R^2_{ad}$ and $RMSE_{ad}$ represent the adjusted $R^2$ and $RMSE$, respectively. $R^2_{tr}$,

$R^2_{va}$, and $R^2_{te}$ are the $R^2$ values of the training, validation, and testing sets, respectively; $RMSE_{tr}$, $RMSE_{va}$,

and $RMSE_{te}$ are the $RMSE$ values of the training, validation, and testing sets, respectively. The ML model

with the highest $R^2_{ad}$ and lowest $RMSE_{ad}$ is regarded as the optimal model for each season in Fig. 1b.

(5) Gridded rice yield generation

For each case, predictors of gridded scale consistent with administrative scale (Sect. 2.3.2) were input

into the optimal model and the gridded rice yield was generated from 1995 to 2015. All the27 cases

followed this process and were combined to generated the AsiaRiceYield4km dataset.

(6) Uncertainty spatialization

To provide the spatial uncertainty, the relative $RMSE$ ($RRMSE$, Eq. 8) of AsiaRiceYield4km was

calculated according to (Luo et al., 2020b). $RRMSE$ of each administrative unit was allocated to the

centroid of the unit and kriging interpolation method was used to spatialize the distribution of uncertainty.

$$RRMSE = \sqrt{\sum_{i=1}^{m} \left( \left( Y_{i,j}^{es} - Y_{i,j}^{ob} \right) / Y_{i,j}^{ob} \right)^2 / m} \cdot 100\% \tag{8}$$

where $m$ is the total number of the year.

**3 Results**

**3.1 Performance of the estimated models**

After selecting the optimal ML model for each case, we scattered the seasonal training, validation, testing,

and adjusted accuracy in Fig. 3. The training $R^2$ is higher than 0.9 for all cases, followed by validation

and testing $R^2$ (average: 0.78 and 0.69, respectively). The $R^2_{ad}$ ranges from 0.60 to 0.90 (average: 0.77),

with the lowest $R^2_{ad}$ in the single season of Malaysia and the highest $R^2_{ad}$ in the winter season of

Bangladesh (Fig. 3c). As for $RMSE$, the averages for training, validation, and testing are 105, 408, and

489 kg/ha, respectively. The $RMSE_{ad}$ ranges from 162 to 817 kg/ha and its average is 396 kg/ha. The

highest $RMSE_{ad}$ is for single rice in China (Fig. 3d). The rice yields of China are mostly higher than those

of other countries, which might cause more modeling uncertainties. For double rice systems (Fig. 3b and 3e), there is no significant difference between their modeling accuracies, with approximately 0.77 for $R_{ad}^2$ and 410 kg/ha for $RMSE_{ad}$. For triple rice, the winter season in Bangladesh has the highest $R_{ad}^2$ (0.90; No. 24 dot in Fig. 3c), and the spring season in Vietnam has the lowest $RMSE_{ad}$ (327 kg/ha; No. 25 dot in Fig. 3c). Additionally, 27 optimal models consist of two types of ML models—XGBoost for 15 seasons and RF for 12 seasons—with no LSTM model. The 27 optimal models and their hyper-parameters are listed in Table S5.

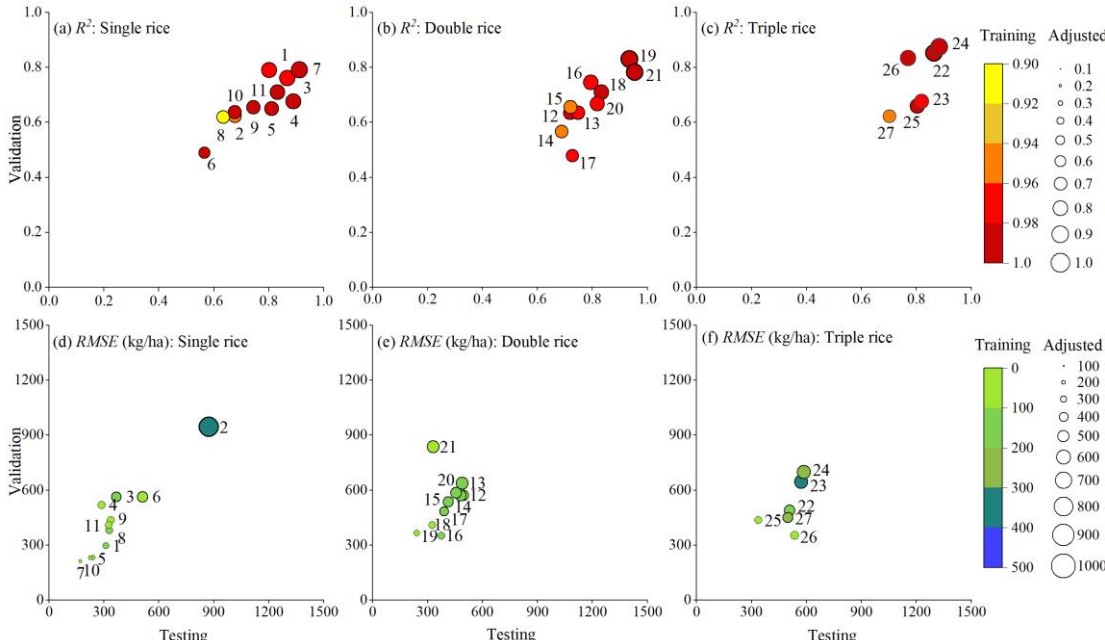

**Figure 3: Accuracy ($R^2$ a-c and $RMSE$ d-f) of the estimated yields for seasonal rice in each region. The $R^2$ (a-c) and $RMSE$ (d-f) are presented in the top panel and bottom panel, respectively. The color of the dots indicates different training accuracy ranks; testing accuracy on the x-axis; validation accuracy on the y-axis; and the size of dots represents the adjusted accuracy. Note: numbers for each dot represent each case shown in Fig. 1b.**

**3.2 Comparing AsiaRiceYield4km products with the observations**

After aggregating AsiaRiceYield4km into administrative units, we compared them with the observed yield at administrative and annual scales. At the administrative scale, comparisons were separately conducted for single, double, and triple rice, as shown in Fig. 4. The estimated and observed yields are closed around the 1:1 line. The overall $R^2$ is higher than 0.87, while the $RMSE$ is lower than 921 kg/ha, suggesting that AsiaRiceYield4km is mostly identical to the observations. The accuracy of single rice ($R^2$: 0.88 and $RMSE$: 920 kg/ha) is slightly lower than that of double rice ($R^2$: 0.91 and $RMSE$: 554 kg/ha)

and triple rice ($R^2$: 0.93 and $RMSE$: 494 kg/ha), mainly because some high-yielding units are not well estimated for single rice (Fig. 4a). Moreover, late rice shows higher accuracy than early rice ($R^2$: 0.92 > 0.89, $RMSE$: 553 kg/ha < 556 kg/ha), which is consistent with the previous study (Cao et al., 2021). As

for triple rice, winter rice has higher accuracy than spring and autumn rice even though its yield range was the greatest.

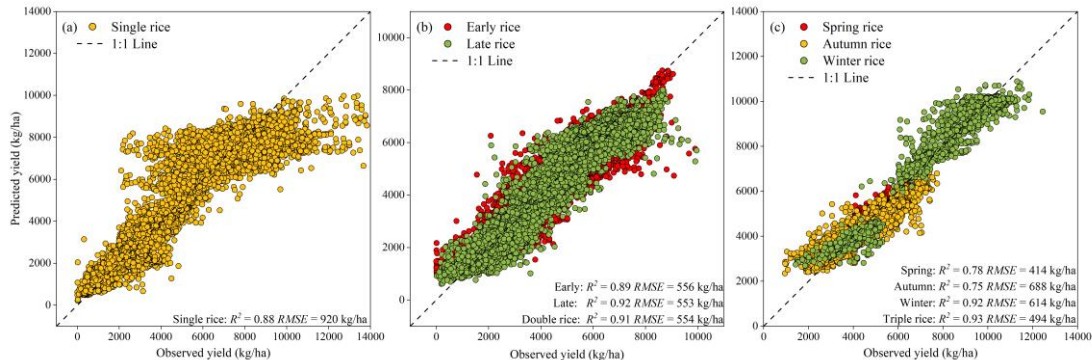

**Figure 4: Comparison of AsiaRiceYield4km with observed yields at administrative units for (a) single rice, (b) double rice, and (c) triple rice.**

At the interannual scale, annual average yield of AsiaRiceYield4km and observed yields for each case are presented. All seasons are statistically highly significant ($p < 0.001$), and $R^2$ of all the results is higher than 0.8. In addition, the differences of SD are also presented in Fig. 5. The largest difference is the early season for double rice in Vietnam which is mainly attributed to the underestimation of AsiaRiceYield4km after 2006. All differences of std are lower than 200 kg/ha, indicating that

AsiaRiceYield4km can well estimate and capture the interannual variations in observed yields.

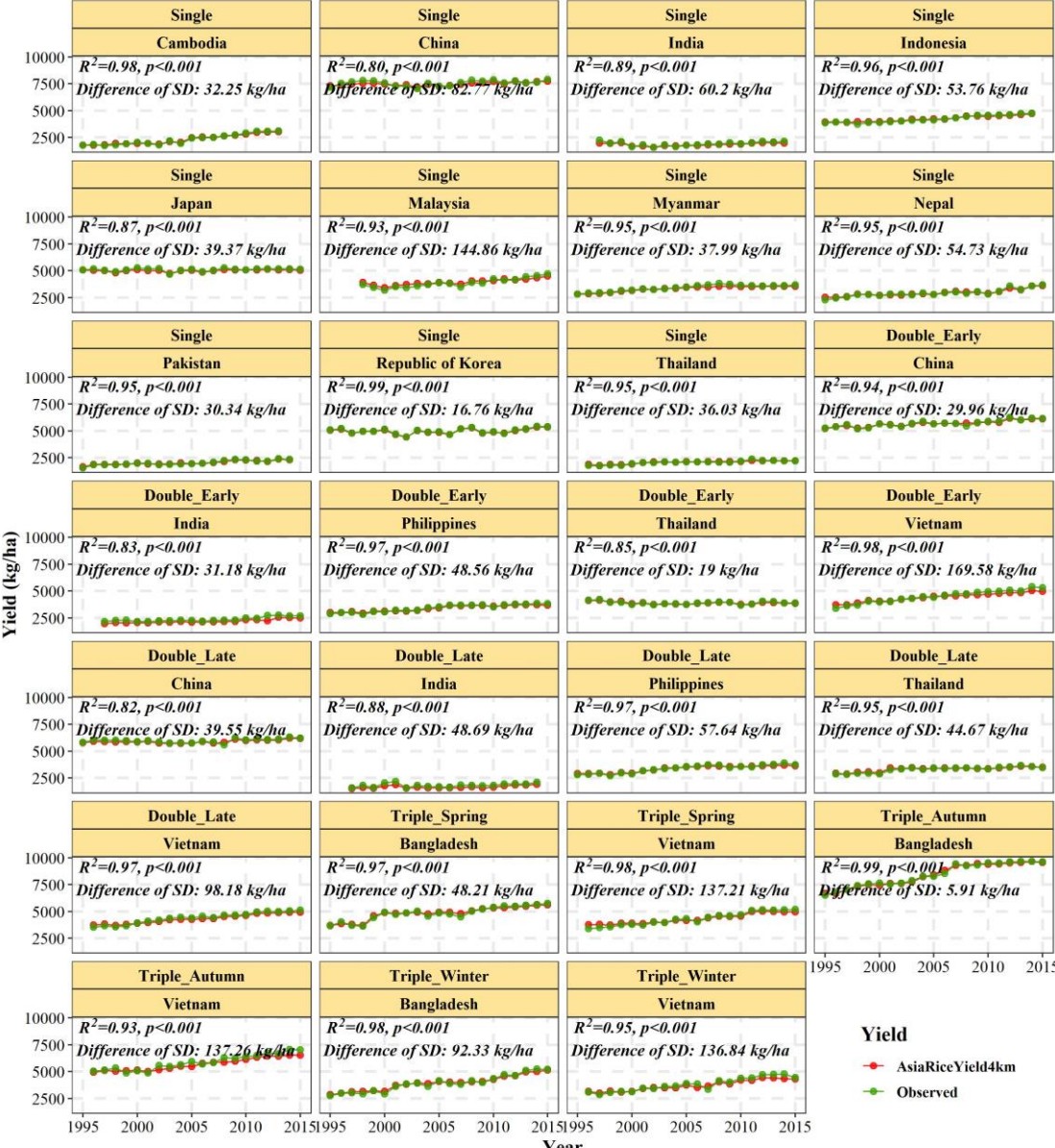

**Figure 5: Interannual comparison of AsiaRiceYield4km with observed yield from 1995 to 2015.**

### 3.3 Comparing AsiaRiceYield4km products with SPAM

Due to the limited time coverage and rice season information of SPAM, only single rice in 2000, 2005,

and 2010 were compared between AsiaRiceYield4km and SPAM. The spatial distribution of rice yield

for AsiaRiceYield4km, SPAM, and observed yield in 2005 are presented in Fig. 6a-c with the zoom-in

views of the Indo - Gangetic Plain (IGP) in Pakistan and India (Fig. 6 a1-c1). After aggregating

AsiaRiceYield4km and SPAM to administrative units, both products were also quantitatively compared

with the observed yield in Fig. 6d for 2005. Similar comparisons for 2000 and 2010 are shown in Fig.

S1. Compared with SPAM, AsiaRiceYield4km has a higher $R^2$ and a lower $RMSE$. Specifically, the $R^2$

of AsiaRiceYield4km is 0.18, 0.23, and 0.20 higher and the corresponding *RMSE* value is 570, 692, and

592 kg/ha lower, respectively, than that of SPAM in 2000, 2005, and 2010. Moreover,

AsiaRiceYield4km shows better spatial consistency with the observed yield across the whole area. The

yield spatial variation in AsiaRiceYield4km and the observed yield are identical in the IGP, while some

360 administrative unit yields of SPAM are overestimated (Fig. 6a1-c1).

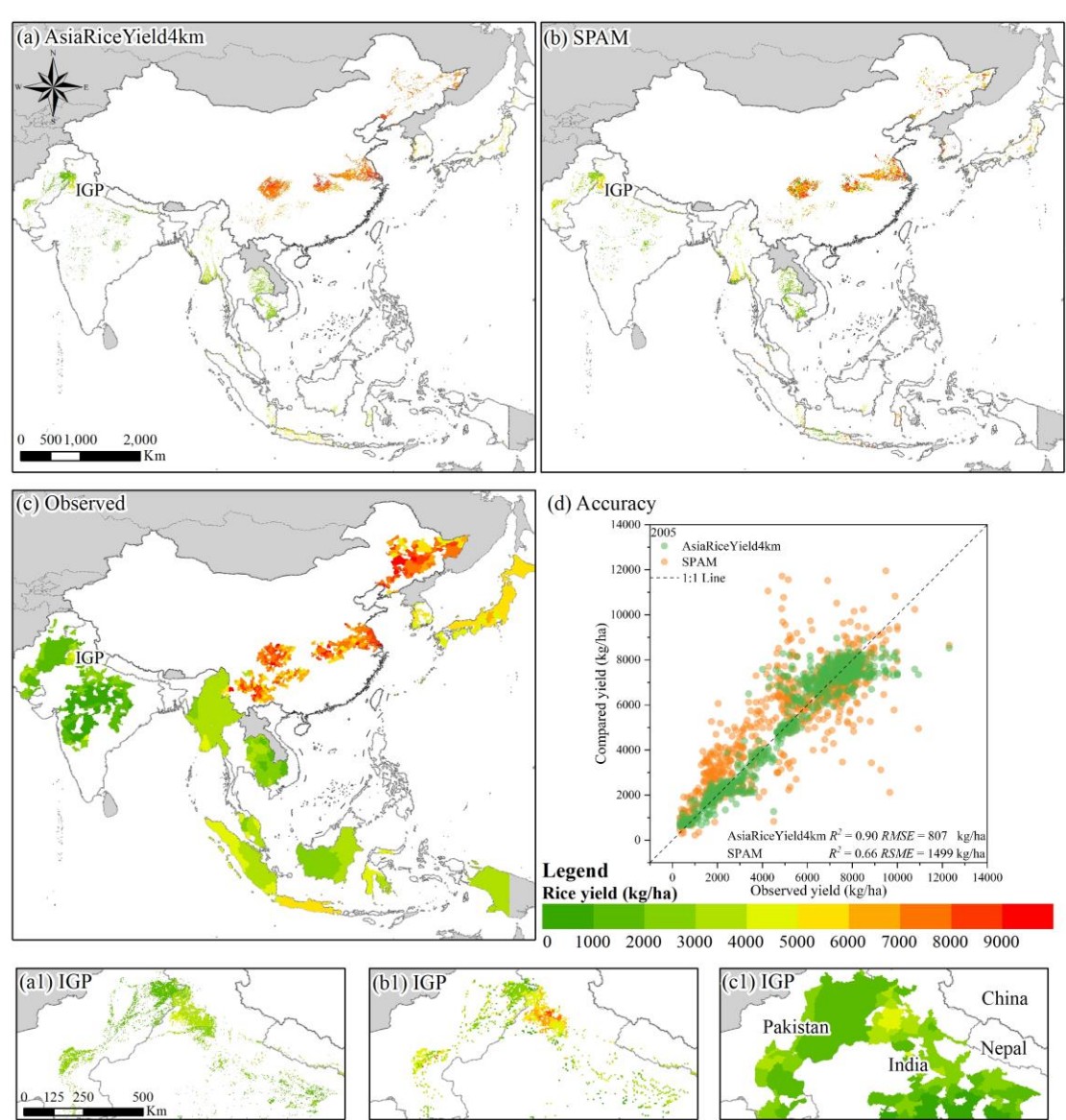

**Figure 6: Yield distribution of (a) AsiaRiceYield4km, (b) SPAM, and (c) observed yields in 2005, and (d)**
**quantitative comparisons with the observed yields in 2005. (a1) to (c1) are the zoom-in views of the IGP in**
**Pakistan and India, with (a1) for AsiaRiceYield4km, (b1) for SPAM, and (c1) for the observed yields.**

**3.4 Spatiotemporal characterizations of AsiaRiceYield4km**

Based on the estimated seasonal yields from optimal ML models, we characterized the spatiotemporal

patterns of rice yields during the period 1995-2015. At the spatial scale, single rice is widely distributed

in 11 countries across the whole area, where its yield varies greatly from 400 to 10000 kg/ha with an average of 5428 kg/ha. Specifically, the highest average yield is in China (7384 kg/ha) and the lowest

yield is in India (1889 kg/ha). Such a large difference might be ascribed to better irrigation in China (Dawe et al., 2010) and relatively low-level soil fertility, investment, and technology in India (Srivastava and Mahapatra, 2012). Double rice mostly distributed between 30°N~0°. Double rice has insignificant differences between early yield and late yield: early rice ranges from 1041 to 8347 kg/ha with an average yield of 4598 kg/ha; late rice ranges from 666 to 7977 kg/ha with an average yield of 4539 kg/ha. Triple

rice seasons are planted in Bangladesh and Vietnam. The ranges of rice yield for spring, autumn, and winter are from 3034 to 6249, from 2690 to 6986, and from 2514 to 10870 kg/ha, with corresponding averages of 4153, 4716, and 7794 kg/ha, respectively. Notably, the highest average yield is 8597 kg/ha for winter rice in Bangladesh, due to its high-yielding hybrid varieties and well-managed fieldwork (e.g., fully irrigated increasing fertilizer, pesticides, and herbicides applications) (Meroni et al., 2021).

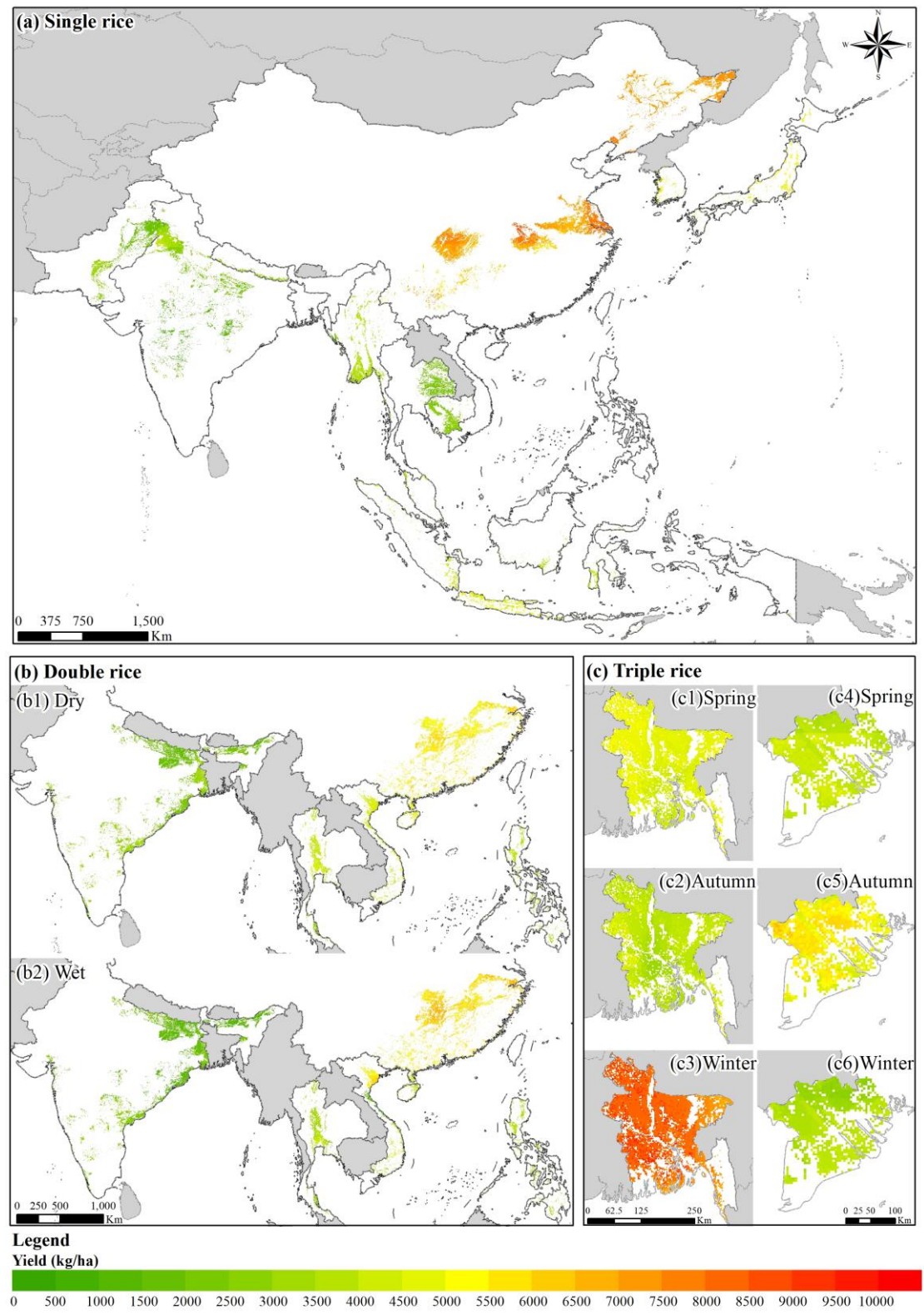

**Figure 7: Spatial patterns of the estimated rice yields (averages during the period 1995-2015) for different seasons.**

For the temporal scale, the interannual rate of yield change (defined as yield difference of last year and current year divided by yield of last year) from 1995 to 2015 for each case is shown in Fig. 8. The annual rate ranges from -18.55% to 25.57%. The average interannual rate during 1995-2015 increases for most cases, with the exception of single rice for Japan (-0.01%) and the early season of double rice for Thailand (-0.11%). Among all cases, the greatest average rate is 2.65% in Cambodia.

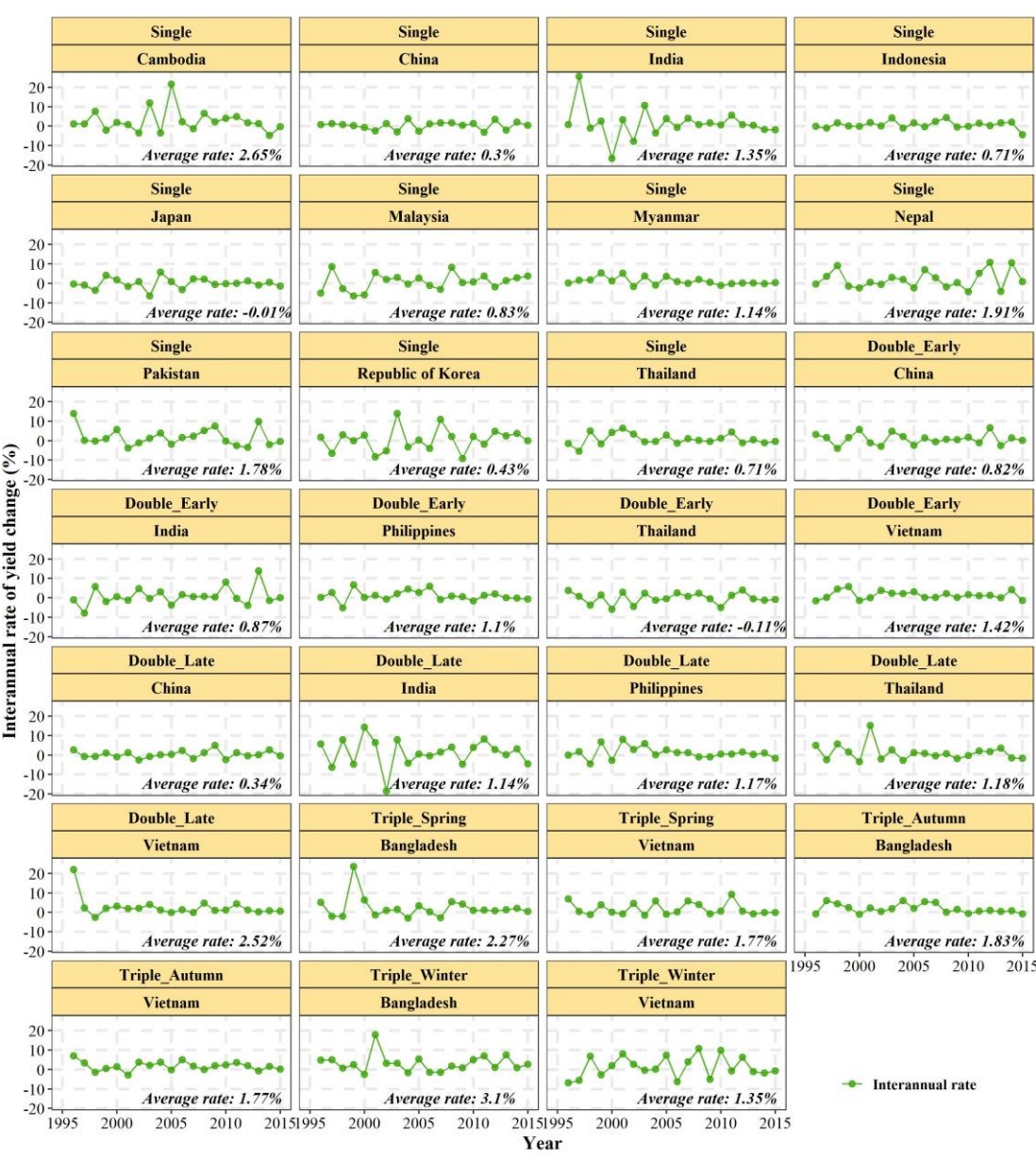

**Figure 8: Temporal variation in the estimated rice yield change for different seasons from 1995 to 2015.**

 **4 Discussion**

**4.1 Frequency and importance of the predictors in ML models**

In this study, 50 predictors were used in ML models but their contributions greatly varied. First, only predictors having a significant correlation with yields were selected for ML models, with the exception of temporal and spatial predictors (*Year*, *Lon*, *Lat*, and *Ele*) (details in Sect. 2.3.2). As a result, the
selection frequency of temporal and spatial predictors was 27 times and the selection frequency of other predictors ranged from 2 to 25 times (Fig. 9a). Using the selected predictors, ML models then estimated rice yields and ranked the importance of each predictor (Fig. 9a). The results showed that temporal and spatial predictors had relatively greater average importance ($>0.05$) and that the importance of the remaining predictors was lower than 0.03 (Fig. 9a).

For different growing periods, REP predictors had greater average importance (0.010) in ML models followed by WGP and VEP predictors (0.007 and 0.005). The average selection frequency for WGP and VEP predictors (8.4 and 10.9 times, respectively) was much lower than that of REP (14.5 times). Therefore, REP predictors contributed the most to yield estimation, which was also consistent with previous studies (Chang et al., 2005; Nazir et al., 2021). In addition, we also found that EGP
predictors (0.014 and 21.3 times) had greater average importance and selection frequency than CGP predictors (0.007 and 11.3 times), respectively, indicating the stronger response of rice yields to extreme growth conditions.

Figure 9b further proportioned the importance of the four predictor categories for each seasonal rice. Although the proportioned importance varied for different rice seasons, the overall contribution was
highest for CEC predictors (45%) followed by EGP (21%), TI (18%), and CGP predictors (16%). CEC had the greatest proportioned importance for most countries which suggested the great importance of the geographical environment for rice yield estimation. More interestingly, the importance of CEC predictors for Myanmar, Thailand, and the late season of Vietnam exceeded 0.8.

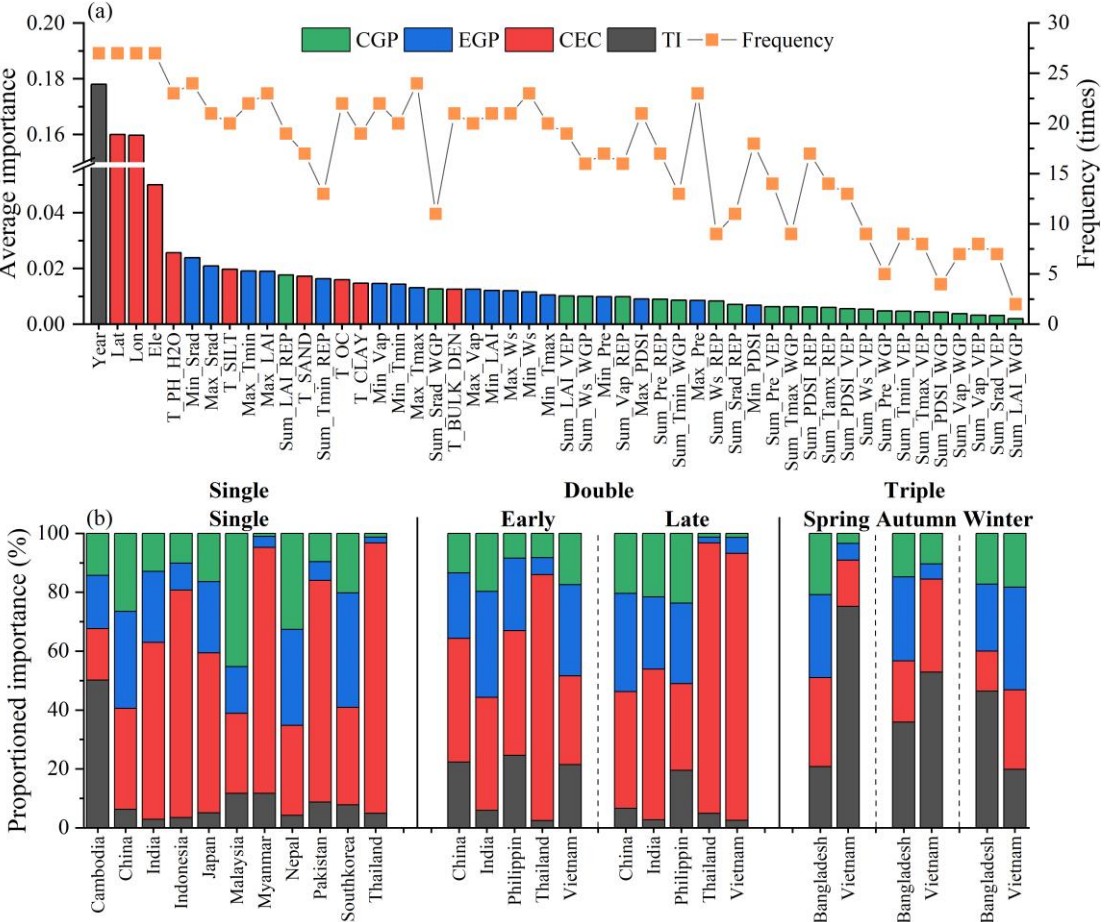

**Figure 9: (a) Average frequency and importance of each predictor. (b) The proportioned importance of each predictor category for seasonal rice.**

## 4.2 Improvements in AsiaRiceYield4km

AsiaRiceYield4km is a seasonal rice yield product with high spatiotemporal resolution and a long time span across the dynamic rice planting areas in the main rice-producing countries of Asia. Compared with SPAM, the spatial resolution of our AsiaRiceYield4km is 4km which is the current highest resolution among all rice yield datasets. Additionally, the product period covers from 1995 to 2015 and includes multi-seasonal rice yields within one year, with more information than most other rice yield datasets. Similarly, AsiaRiceYield4km considered both the annual dynamic change in rice-planting areas and phenological information at a grid scale, rather than a constant planting area map and fixed growing period. Such dynamic information assisted us in capturing better spatial and temporal variations in rice yields, and consequently greatly improved the accuracy of our product. Moreover, we applied four predictor categories and the optimal ML models to estimate seasonal yields. Four predictor categories provided comprehensive rice growth information to ensure the accuracy of yield estimations. The optimal

models for each rice season are determined by the IPW method. As a weighted ensemble assessment to

fully consider training, validation, and testing accuracy, we are certain that the IPW method is more

robust and reasonable to select the optimal model for seasonal rice yield in Asia.

**4.3 Uncertainty analysis**

For the spatial uncertainty, the *RRMSE* values in most areas were below 30%, indicating the low

uncertainty of AisaRiceYield4km. High uncertainty of *RRMSE* (above 50%) was distributed in

northeastern China and western India for single rice and central Bangladesh for winter season of triple

rice (Fig. 10).

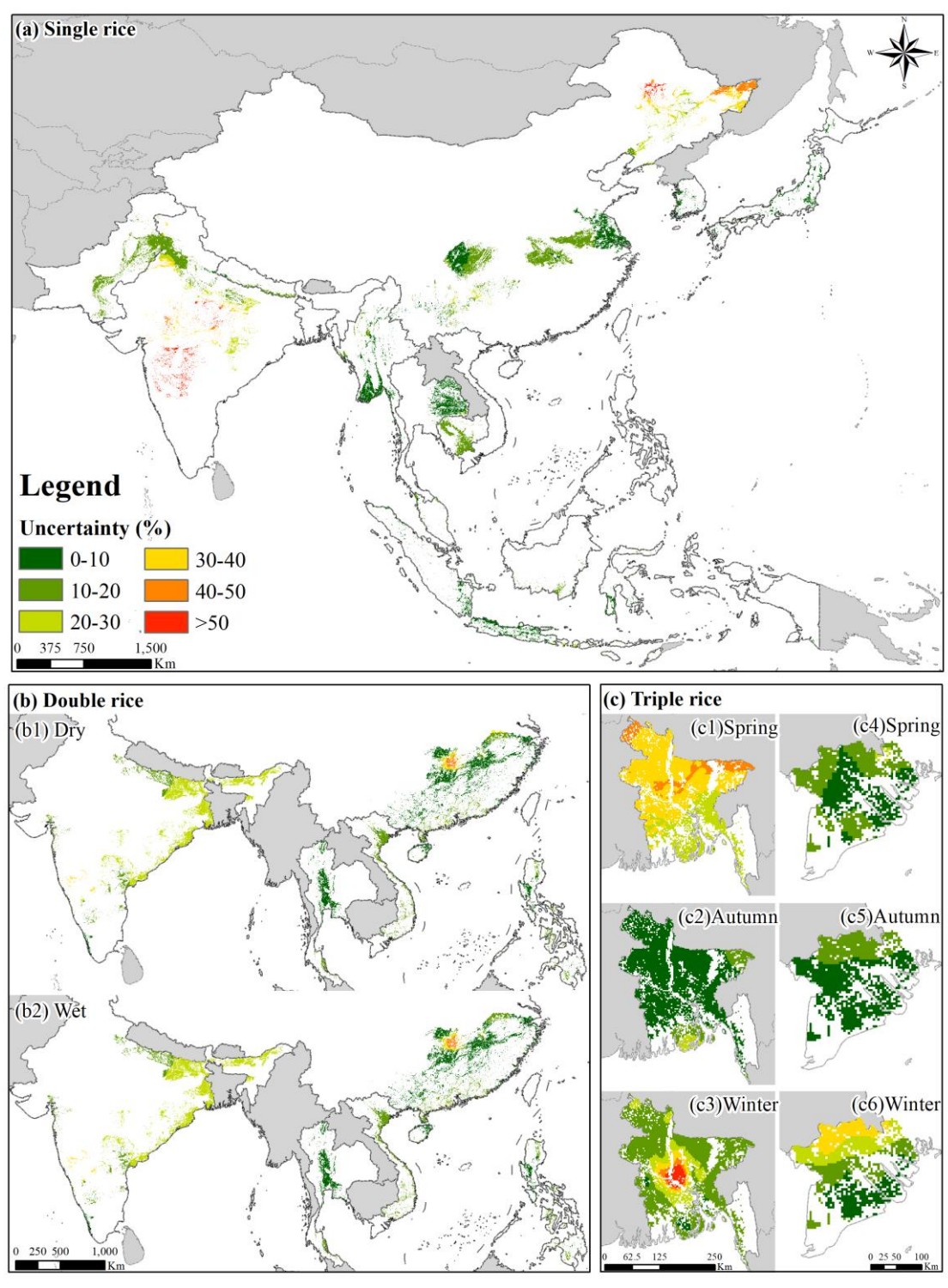

**Figure 10: Spatial distribution of uncertainty (*RRMSE*, %) in AsiaRiceYield4km.**

In this study, we have improved the yield prediction processes to ensure the accuracy of the AsiaRiceYield4km product as much as possible, however, several factors might negatively impact its accuracy. Due to the limitations of remote sensing techniques (e.g., clouds and topography), some paddy rice areas cannot be recognized, consequently leading to map errors (Han et al., 2022). Besides, the rice

areas before 20000 based on the union rice area of 2000 – 2002 was also introduces some uncertainty due to the unavailable of these rice areas. The spatial resolutions of multi-source data also cause uncertainties. For example, given that the rice planting areas in Asia are always fragmented (Lowder et al., 2016) but the LAI resolution in this study is somehow coarser (0.05°), the mixed-pixel problem will inevitably influence the accuracy of AsiaRiceYield4km in small size rice-planting areas. Although the GLASS LAI has highest accuracy and lowest uncertainty and we have made several efforts to mitigate the uncertainty, there is still uncertainty and inevitable effects on the rice yield estimation (Liu et al., 2018; Li et al., 2018; Fang et al., 2019; Chen et al., 2020). In addition, the crop intensity used in this study is administrative scale. The annual crop intensity variation in rice still inflects the yield estimation results. Finally, due to the lack of a process-based mechanism, ML is weakly traceable and interpretable for rice yield variability (Muruganantham et al., 2022), especially for extreme rice yields. Nevertheless, compared with other public products (Fig. 6), our methods still generated better seasonal rice yield predictions at a higher-spatiotemporal-resolution for a longer period.

## 5 Data availability

The seasonal rice yield product for Asia during the period 1995-2015 (AsiaRiceYield4km) is available at https://doi.org/10.5281/zenodo.6901968 (Wu et al., 2022). We encourage users to independently verify data products before using them.

## 6 Conclusions

We produced a long-term seasonal rice yield dataset with high spatiotemporal resolution on dynamic paddy rice areas in Asia by using multi-source data and ML models. Our AsiaRiceYield4km dataset has higher accuracy than other public datasets and shows more spatial consistency with the observed yield. We attributed such improvements to more dynamic information (e.g., rice area and phenological dates), full consideration of rice growth conditions, and the novel IPW method to select the optimal ML model. Moreover, we discovered that constant environmental conditions contributed the most (~45%) to rice yield prediction than other growing conditions. Predictors in REP had more impacts on yield predictions than those in WGP and VEP. Our dataset can address the lack of seasonal rice yield datasets and support studies related to agricultural production and development.

**Author contributions.**

Conceptualization, Z.Z and F.T.; Data curation, Y.L. and J.H.; Formal analysis: H.W. and J.Z.; Funding acquisition: J.Z., Z.Z. and F.T.; Investigation: J.C., J.H., H.W., J.Z., L.Z., and Y.L.; Methodology, J.Z. and H.W.; Software, H.W. and J.Z.; Supervision, Z.Z., F.T. and J.X.; Validation, J.Z. and J.H.; Visualization: H.W. and J.Z.; Writing – original draft preparation: H.W. and J.Z.; Writing – review & editing: J.Z., Z.Z., and Q.M. All authors have read and agreed to the published version of the manuscript.

**Competing interests.**

The contact author has declared that neither they nor their co-authors have any competing interests.

**Disclaimer.**

Publisher's note: Copernicus Publications remains neutral with regard to jurisdictional claims in published maps and institutional affiliations.

**Acknowledgments.**

We would like to thank the editors and anonymous reviewers for their valuable comments. And we would like to thank the support of the open project of the Key Laboratory of Environmental Change and Natural Disasters, Ministry of Education, Beijing Normal University.

**Financial support.**

This research was funded by the National Key Research and Development Project of China (2020YFA0608201), China National Postdoctoral Program for Innovative Talents (BX20200064), and National Natural Science Foundation of China (42061144003, 41977405, 42101095).

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
