# Peer review of "AsiaRiceYield4km: Seasonal Rice Yield in Asia from 1995 to 2015"

_Earth System Science Data, 2022_

## Author Comment (AC1)

**Review #1:**

**General Comment:** This manuscript developed a high spatial resolution (4km) rice yield dataset from 1995 to 2015, covering major rice growing seasons and regions in Asia. Overall, this dataset would be a good complement to current rice yield products due to its high spatiotemporal resolution. I have the following questions or suggestions, which may help improve the manuscript clarity.

**Response to general comment:**

We are grateful for anonymous referee #1's recognition of this study's importance. We carefully revised our manuscript and provided a point-by-point response below. We have addressed all points raised in the revised manuscript.
* * *
Note: The individual comments (shown in black) are listed below including our responses (shown in blue) and revised parts in the manuscript (shown in *red and italic font*). Line numbers (shown in **blue and bold font**) that we mention in this comment refer to our revised manuscript with all markup version.

**Comment 1:** The authors used the GLASS AVHRR LAI data to extract key crop phenological indicators for training, including planting, heading, and harvesting dates. However, since rice fields in Asia are very fragmented and the spatial resolution of GLASS LAI data (i.e., 0.05 deg) is not fine enough to capture pure rice LAI information, there should be mixed-pixel problems. How did the authors deal with these problems? In addition, I would say the extracted planting and harvesting dates are more of indicators of the early rapid growth and senescence stages rather the real planting and harvesting dates. The authors should clarify these conceptual differences to avoid possible confusions.

**Response to comment 1:**

Thank you very much for your comments and suggestions.

Yes, mixed-pixed problems could impact accurately retrieving crop information. We agree with you that the problems could affect the extraction of LAI information. Fortunately, some efforts can reduce the mixed-pixel influence in some degree such as our efforts. Firstly, GLASS LAI product has the highest accuracy and the lowest uncertainty compared with other available LAI products (Xiao et al., 2016; Liang et al., 2021). Secondly, we used annual paddy rice of 500 m as base maps which can reduce the influence of other land cover types by capturing the dynamic temporal variation of rice distribution (**Lines 124-125**). Moreover, only pixels with LAI value within or equal to average $\pm$ two times standard deviation were selected to identify rice growing information for the reduction the interference of abnormal values (**Lines 211-212**). Finally, we filtered out a fraction of pixels where the rice growing information couldn't be detected by inflection-based and threshold-based methods (details in Sect. 2.3.1). These measures helped us to reduce the influence caused by mixed-pixel problems. The accuracy of phenological information used in this study was satisfactory enough ($R^2 > 0.8$) for the main rice-cropping seasons according to Zhang et al. (2022). Nevertheless, we further discussed the relevant uncertainties in Sect. 4.3.1 (**Lines 436-439**).

Thank you for pointing out these conceptual differences of the phenological information. The extracted planting dates were the transplanting dates which located in the early rapid growth stage (Mandal et al., 2018). For harvesting dates, they referred to the occurrence of leaf senescence at maturity period (Ogawa et al., 2021; Ni et al., 2021; Zhou et al., 2019). These two dates are truly indicated the early rapid growth and senescence stages. However, these extraction rules were thought as transplanting

and maturity dates detection according to most previous studies (Luo et al., 2020; Niu et al., 2022). To avoid ambiguity, we replaced planting with transplanting and harvesting with maturity according to relevant researches (Dong and Xiao, 2016;). Correspondingly, the figure of LAI extraction in Fig. 2 Step1 was also revised.

References:

Dong, J. and Xiao, X.: Evolution of regional to global paddy rice mapping methods: A review, ISPRS J. Photogramm. Remote Sens., 119, 214–227, https://doi.org/10.1016/j.isprsjprs.2016.05.010, 2016.

Liang, S., Cheng, J., Jia, K., Jiang, B., Liu, Q., Xiao, Z., Yao, Y., Yuan, W., Zhang, X., and Zhao, X.: The global land surface satellite (GLASS) product suite, Bull. Am. Meteorol. Soc., 102, E323–E337, https://doi.org/10.1175/BAMS-D-18-0341.1, 2021.

Luo, Y., Zhang, Z., Chen, Y., Li, Z., and Tao, F.: ChinaCropPhen1km: a high-resolution crop phenological dataset for three staple crops in China during 2000–2015 based on leaf area index (LAI) products, Earth Syst. Sci. Data, 12, 197–214, https://doi.org/10.5194/essd-12-197-2020, 2020.

Mandal, D., Kumar, V., Bhattacharya, A., Rao, Y. S., Siqueira, P., and Bera, S.: Sen4Rice: A processing chain for differentiating early and late transplanted rice using time-series Sentinel-1 SAR data with Google Earth engine, IEEE Geosci. Remote Sens. Lett., 15, 1947–1951, 2018.

Ni, R., Tian, J., Li, X., Yin, D., Li, J., Gong, H., Zhang, J., Zhu, L., and Wu, D.: An enhanced pixel-based phenological feature for accurate paddy rice mapping with Sentinel-2 imagery in Google Earth Engine, ISPRS J. Photogramm. Remote Sens., 178, 282–296, 2021.

Niu, Q., Li, X., Huang, J., Huang, H., Huang, X., Su, W., and Yuan, W.: A 30-m annual maize phenology dataset from 1985 to 2020 in China, Earth Syst. Sci. Data Discuss., 1–28, 2022.

Ogawa, D., Sakamoto, T., Tsunematsu, H., Kanno, N., Nonoue, Y., and Yonemaru, J.: Remote-Sensing-Combined Haplotype Analysis Using Multi-Parental Advanced Generation Inter-Cross Lines Reveals Phenology QTLs for Canopy Height in Rice, Front. Plant Sci., 12, 2021.

Xiao, Z., Liang, S., Wang, J., Xiang, Y., Zhao, X., and Song, J.: Long-time-series global land surface satellite leaf area index product derived from MODIS and AVHRR surface reflectance, IEEE Trans. Geosci. Remote Sens., 54, 5301–5318, https://doi.org/10.1109/TGRS.2016.2560522, 2016.

Zhang, J., Wu, H., Zhang, Z., Zhang, L., Luo, Y., Han, J., and Tao, F.: Asian Rice Calendar Dynamics Detected by Remote Sensing and Their Climate Drivers, Remote Sens., 13, https://doi.org/10.3390/rs14174189, 2022.

Zhou, G., Liu, X., and Liu, M.: Assimilating remote sensing phenological information into the WOFOST model for rice growth simulation, Remote Sens., 11, 268, 2019.

**Comment 2:** The authors used the Pearson correlation analysis to identify those predictors with a significant correlation with rice yield at each administrative unit for training (Lines 218-220). I'm curious if the authors trained the model in each administrative unit and then combined all the training results to get the rice yields for the entire Asian region. More explanations about the experimental implementations should be given. Meanwhile, how do the authors deal with the multicollinearity problems of these predictors? There is a significant correlation between the different predictors in Table S3. In addition, I found very limited information on hyper-parameters in the supplementary material, the authors may want to provide detailed information of those parameters in each optimal model (e.g., how many hidden layers, node numbers, and max-depth, etc). Furthermore, in Line 295, detailed information on the trained 27 optimal models should also be give (maybe present in the supplementary material).

**Response to comment 2:**

Thanks very much for your constructive comment.

We trained the optimal models in each case (one specific rice-cropping period, including all administrative units in the country. Such training case contains many administrative units which are at the minimum administrative division scale with available rice yield records from 1995 to 2015. The gridded predictors selected in these cases were input into the optimal models to produce the gridded rice yield and all the gridded rice yield were combined to get the AsiaRiceYield4km dataset. We agreed with you that more experimental implementations should be given, thus we added more details in the revised manuscript Sect. 2.3.2 **(Lines 226-229)** and one new paragraph named *(5) Gridded rice yield estimation* in Sect. 2.3.3 **(Lines 296-299)**. Besides, Figure 2 was adjusted correspondingly.

Multicollinearity problems can affect the performance of regression models (Ma and Cheng, 2016; Yang et al., 2022), but machine learning (ML) can overcome this problem in some degree (Feng et al., 2016; Zhao et al., 2019; Guo et al., 2021; Chan et al., 2022). ML can capture non-linear relationships and handle the interactions among predictors (Breiman, 2001; Shalev-Shwartz and Ben-David, 2014; Leng and Hall, 2020). Specifically, both random forest (RF) and extreme gradient boosting (XGBoost) are tree-based algorithms which can inherently immune to multicollinearity problems (Guo et al., 2021). Besides, the bagging process in RF and bootstrapping process in XGBoost can also mitigate multicollinearity effects according to Ma and Cheng (2016) and Ma (2020).

For hyper-parameters, we followed your suggestions to add hyper-parameter spaces in the revised supplement. Besides, more details about the defined space and the optimal set of values were listed in the supplement (Table S4 and S5) and the Python library details of ML algorithms were also presented **(Lines 34-36 and 37-39 in the supplement)**.

References:

Breiman, L.: Random forests, Mach. Learn., 45, 5–32, 2001.

Chan, J. Y.-L., Leow, S. M. H., Bea, K. T., Cheng, W. K., Phoong, S. W., Hong, Z.-W., and Chen, Y.-L.: Mitigating the Multicollinearity Problem and Its Machine Learning Approach: A Review, Mathematics, 10, 1283, 2022.

Feng, G., Mao, L., Sandel, B., Swenson, N. G., and Svenning, J.-C.: High plant endemism in China is partially linked to reduced glacial-interglacial climate change, J. Biogeogr., 43, 145–154, 2016.

Guo, M., Yuan, Z., Janson, B., Peng, Y., Yang, Y., and Wang, W.: Older pedestrian traffic crashes severity analysis based on an emerging machine learning XGBoost, Sustainability, 13, 926, 2021.

Leng, G. and Hall, J. W.: Predicting spatial and temporal variability in crop yields: an inter-comparison of machine learning, regression and process-based models, Environ. Res. Lett., 15, 044027, 2020.

Ma, J. and Cheng, J. C.: Identifying the influential features on the regional energy use intensity of residential buildings based on Random Forests, Appl. Energy, 183, 193–201, 2016.

Ma, J., Ding, Y., Cheng, J. C., Jiang, F., Tan, Y., Gan, V. J., and Wan, Z.: Identification of high impact factors of air quality on a national scale using big data and machine learning techniques, J. Clean. Prod., 244, 118955, 2020.

Shalev-Shwartz, S. and Ben-David, S.: Understanding machine learning: From theory to algorithms, Cambridge university press, 2014.

Yang, B., Xiao, Z., Meng, Q., Yuan, Y., Wang, W., Wang, H., Wang, Y., and Feng, X.: Deep learning-based prediction of effluent quality of a constructed wetland, Environ. Sci. Ecotechnology, 100207, 2022.

Zhao, X., Yu, B., Liu, Y., Chen, Z., Li, Q., Wang, C., and Wu, J.: Estimation of poverty using random forest regression with multi-source data: A case study in Bangladesh, Remote Sens., 11, 375, 2019.

**Comment 3:** The authors compared their dataset with observations via scatter plots (Figure 5). This is good. However, it would be better if the authors can additionally provide comparisons of the interannual variations in rice yield for each rice system (e.g., single, double early and later) in each country (there should be some survey data). The performance of your dataset in capturing interannual variations in rice yield is important.

**Response to comment 3:**

Thanks very much for your constructive comment.

We agree with you that the comparison of interannual variation is essentially important for rice yield dataset. Here, we added interannual comparison between AsiaRiceYield4km and observed yields for all countries. The results showed that our dataset has good consistency with the observed yield for all rice growing seasons. This comparison result analysis was added to the revised *Section 3.2 Comparing AsiaRiceYield4km products* in the manuscript (**Lines 338 to 345**).

**Comment 4.** The authors used cumulative values of predictors (e.g., LAI and PDSI) in different phenological periods (e.g., vegetative and reproductive) to train models. However, these cumulative information has no actual physiological significance. Meanwhile, considering that crop phenological dates (e.g., planting and harvesting) vary from year to year, it would be better to use the average value of these predictors over each phenological periods for training (i.e., more comparable across years).

**Response to comment 4:**

We did select eight cumulative growing predictors (CGP) during annual rice phenological stage, including leaf area index (LAI) and seven climate variables. LAI can indicate the vegetational variation in rice growing status and biomass. Therefore, we believe the cumulated LAI predictors have the actual physiological significance as many previous studies confirmed. Meanwhile, the cumulative climate predictors represent weather conditions during rice growing period which have no physiological significance, the same for the cumulative climate ones.

Nevertheless, we still followed you to replace predictors from CGP category with average values in some cases to validate the estimate results. According to Fig. C1, difference of $R^2$ and *RMSE* for the three cases is 0-0.2 and 7-59 kg/ha, respectively. The predictors from average values had the similar impact on rice yield estimation with those from CGP. Such comparison results are attributed to the good consistency between CGPs and their related averages. Moreover, compared with the monthly resolution of weather predictors, the small change ($\pm 10$ days per decade, Zhang et al., 2022) of temporal variation of rice phenological dates do not significantly affect the results. Therefore, we still used cumulative values for rice yield prediction.

[Figure]

**Fig. C1: The accuracy of AsiaRiceYield4km and the average.**

Reference:

Zhang, J., Wu, H., Zhang, Z., Zhang, L., Luo, Y., Han, J., and Tao, F.: Asian Rice Calendar Dynamics Detected by Remote Sensing and Their Climate Drivers, Remote Sens., 13, https://doi.org/10.3390/rs14174189, 2022.

**Comment 5:** I would suggest that the authors get editing help from someone with full professional proficiency in English, as the current manuscript has substantial language issues. I pointed out some, but not all.

**Response to comment 5:**

Thanks very much for your suggestions.

The manuscript was carefully revised with the help of professional editors of AJE (https://www.aje.cn/?_ga=2.249467463.1174155384.1668480853-862469041.1668480853, last accessed: 15 November 2022). The editing certificate was as follows:

[Figure]

**Figure C2: Editing certificate for the manuscript.**

Other concerns:

**Comment 6:** Line 72: When you say prediction, it is more of a future period than a historical period.

**Response to comment 6:**

Thank you. We realized that it is inappropriate to use rice yield prediction for a historical period dataset. We change "prediction" to "estimation" and "predicted" to "estimated" throughout the manuscript.

**Comment 7:** Line 112: Change "i.e., " to "e.g., "

**Response to comment 7:**

Corrected as suggested. The same errors were also corrected in **Line 434**.

**Comment 8:** Line 113: Change "Philippines" to "China": the season number of 12 and 13 should belong to China.

**Response to comment 8:**

Thank you, we apologized for our carelessness. We have made this correction to the manuscript.

**Comment 9:** Line 117: Change "are" to "were".

**Response to comment 9:**

Corrected as suggested.

**Comment 10:** Line 275: Have you tried any other proportions (e.g., 0.6/0.2/0.2) to examine the robustness of your datasets, trained models and evaluation results?

**Response to comment 10:**

According to your suggestion, we have tried different proportion strategies for ML models (Table C1). For the two dataset division strategies, we used $R^2$ and $RMSE$ of training, validation, testing and estimation result for accuracy comparison. For the two division strategies, the results showed similar accuracy. It suggested that our datasets, trained models and evaluation results were robustness.

**Table C1: Accuracy of rice yield estimation for different proportion strategies.**

| Case | Division strategy | $R^2$ (%) | | | | RSME (kg/ha) | | | |
|------|-------------------|-----------|---|---|---|--------------|---|---|---|
| | | Training | Validation | Testing | Estimation | Training | Validation | Testing | Estimation |
| Single season for Republic of Korea | 0.6/0.2/0.2 | 99 | 69 | 67 | 79 | 22 | 232 | 219 | 190 |
| | 0.56/0.24/0.2 | 99 | 68 | 64 | 80 | 25 | 226 | 232 | 186 |
| Early season for Thailand | 0.6/0.2/0.2 | 99 | 83 | 70 | 85 | 37 | 322 | 412 | 303 |
| | 0.56/0.24/0.2 | 99 | 83 | 71 | 84 | 39 | 326 | 409 | 314 |
| Autumn season for Vietnam | 0.6/0.2/0.2 | 99 | 77 | 84 | 64 | 53 | 510 | 332 | 633 |
| | 0.56/0.24/0.2 | 99 | 77 | 83 | 65 | 67 | 536 | 353 | 618 |

**Comment 11:** Figure 3: What does the legend mean? I didn't see any difference in the color of these dots.

**Response to comment 11:**

For Fig. 3, the legend referred to the training accuracy ($R^2$ and *RMSE*). We are sorry that the previous legend range is too large ($R^2$: 0 - 1; *RMSE*: 0 - 1000kg/ha), resulting in no differences for estimated models. We have adjusted the legend range to: $R^2$ from 0.9 to 1 and *RMSE* from 0 to 500kg/ha, as the training $R^2$ was over 0.9 and the training *RMSE* was lower than 400 kg/ha for all optimal models (**Line 316-322**).

**Comment 12:** Section 3.2: I would suggest moving this section to the end of "3 Results". Meanwhile, you should add additional analysis of temporal variations.

**Response to comment 12:**

Thanks very much for your constructive comment. We have moved Sect. 3.2 to the end of Sect. 3 and adjusted the title to *3.4 The spatiotemporal spatial characterizations of AsiaRiceYield4km*. The analysis of temporal variations for rice yield was also added (**Lines 381-387**).

**Comment 13:** Line 417: Add using: by "using" multi-source

**Response to comment 13:**

Corrected as suggested.

**Comment 147:** Table S1: names of the local administrative unit presents the specific… -> names of the local administrative unit represent the specific…

**Response to comment 14:**

Corrected as suggested.

**Comment 15:** Table S2: Provide the full names of these abbreviations in the footnotes.

**Response to comment 15:**

The full names have been added.

**Comment 16 :** Table S3: What do you mean in these rows:

      The sum of for whole growing period

      The sum of for vegetative stage

      The sum of for reproductive stage

      The maximum for whole growing period

**Response to comment 16:**

We feel sorry for our carelessness. Variable "wind speed" was missing. Thanks to your kind reminder, we have revised and simplified them in Table S3. These rows were:

Sum of wind speed for whole growing period

Sum of wind speed for vegetative period
Sum of wind speed for reproductive period
Maximum wind speed

Sum of wind speed for vegetative period
Sum of wind speed for reproductive period
Maximum wind speed

---

## Author Comment (AC2)

**Review #2:**

**General Comment:** High-spatial and high-temporal resolution rice yield datasets are lack especially over large regions. The manuscript employed machine learning algorithms to generate long-term high-resolution rice yield over the South Asia, Southeast Asia, and East Asia. Undergoing a study at continental scales like this is a huge project. The 5km rice yield map over the major rice producing countries in Asia from 1995 to 2015 fills the data gap for assessing the impacts of climate change and the sustainable development. However, I have a few major concerns to be addressed so that the manuscript could be more solid.

**Response to general comment:** We are grateful for anonymous referee #2's recognition of this study's importance. We carefully revised our manuscript and provided a point-by-point response below. We have addressed all points raised in the revised manuscript.
* * *
Note: The individual comments (shown in black) are listed below including our responses (shown in blue) and revised parts in the manuscript (shown in *red and italic font*). Line numbers (shown in **blue and bold font**) that we mention in this comment refer to our revised manuscript with all markup version.

**Comment 1:** (1) The rice cultivated area is the fundamental information for rice yield estimation. The manuscript used rice map for each year from 2000 to 2020 while the yield model was developed and used to estimate spatial distribution of rice yield during 1995 to 2015. Since most input dataset used for rice yield model in the study are available for the year 2000 to 2020, why not generating rice yield for 2000 to 2020 so that the map and the rice yield coincided with each other for the same year?

**Response to comment 1:**

Thank you very much for your comments and suggestions.

Our main objective in the study is to produce a long-term rice yield dataset with higher spatiotemporal resolutions and seasonal information across Asia. Although multi-sources data were used, available rice yield was the dominant factor for time span which was essential for model training and accuracy validation. After inputting our most efforts, we can only obtain the yield records of 1995~2015 for most countries (Table S2). Therefore, the time span of this study was selected from 1995 to 2015.

**Comment 2:** (2) Another concern is the way of predictor selection. The authors selected the predictors based on the correlation analysis between indicators and the yield at each administrative unit. While this is in general logic, it might be a problem when great differences existed in cropping patterns and the rice management in an administrative unit. The correlations may fail to achieve a significant level when an improper unit was targeted. This needs more clarification. Please also specify the administrative unit. Is it national level or sub-national level administrative units?

**Response to comment 2:**

Thanks very much for your constructive comment.

The administrative unit is the sub-national level unit which is at the minimum administrative division (including first, second and third levels, Table S1) scale in this study. Differences of the cropping patterns and the rice management do exist at a national level, while those in the minimum administrative division for each country are smaller.

Besides, the selected predictors in our study consistently indicated significant relationships with yield. To make our manuscript clearer, we have added more descriptions for administrative units (**Lines 114-115**). The administrative division for each country were also listed in Table S1 according to your advice.

**Comment 3:** (3) The authors only used one vegetation indicator LAI as the inputs. It is assessed by several research that LAI products are of high uncertainty even for the improved GLASS LAI products. The product still has some abnormal values and unrealistic seasonality especially in winter. From my understanding, using LAI products might introduce high uncertainty in yield model which is unable to be solved.

**Response to comment 3:**

Thanks very much for your constructive comment.

Considering the spatial and temporal resolutions, GLASS LAI products are more appropriate for our research than other latest public products even with higher spatial resolution. GLASS LAI products have the highest accuracy and the lowest uncertainty compared with other available LAI products according to Xiao et al. (2016) and Liang et al. (2021). In addition, the abnormal values and unrealistic seasonality of LAI are always over the northern high latitudes and the equatorial belt due to cloud/snow coverage and low solar zenith angle in winter (Garrigues et al., 2008; Jin et al., 2017). However, for northern areas at high latitudes area, there is no rice planting in winter. Only some rice planting area of Malaysia and Indonesia located in the equatorial belt may be affected by these problems. Moreover, only 5 LAI variables, accounting for one-tenth of all variables, were used and preprocessed to filter abnormal pixels and those without rice growth patterns (Sect. 2.3.1) to reduce the uncertainty from LAI. In the revised manuscript, we have suggested the process for LAI data (**Lines 211-212**) and added the uncertainty of GLASS LAI in section 4.3 (**Lines 439-442**).

References:

Garrigues, S., Lacaze, R., Baret, F., Morisette, J. T., Weiss, M., Nickeson, J. E., Fernandes, R., Plummer, S., Shabanov, N. V., and Myneni, R. B.: Validation and intercomparison of global Leaf Area Index products derived from remote sensing data, J. Geophys. Res. Biogeosciences, 113, 2008.

Jin, H., Li, A., Bian, J., Nan, X., Zhao, W., Zhang, Z., and Yin, G.: Intercomparison and validation of MODIS and GLASS leaf area index (LAI) products over mountain areas: A case study in southwestern China, Int. J. Appl. Earth Obs. Geoinformation, 55, 52–67, https://doi.org/10.1016/j.jag.2016.10.008, 2017.

Liang, S., Cheng, J., Jia, K., Jiang, B., Liu, Q., Xiao, Z., Yao, Y., Yuan, W., Zhang, X., and Zhao, X.: The global land surface satellite (GLASS) product suite, Bull. Am. Meteorol. Soc., 102, E323–E337, https://doi.org/10.1175/BAMS-D-18-0341.1, 2021.

Xiao, Z., Liang, S., Wang, J., Xiang, Y., Zhao, X., and Song, J.: Long-time-series global land surface satellite leaf area index product derived from MODIS and AVHRR surface reflectance, IEEE Trans. Geosci. Remote Sens., 54, 5301–5318, https://doi.org/10.1109/TGRS.2016.2560522, 2016.

**Comment 4:** (4) According to the importance of the indicators, static indicators (Year, Lat, Long, Ele) are much higher than other indicators. For some countries, the proportioned importance of CEC+TI indicators could be higher than 90%. And for the whole study area, the CEC+TI are the most important indicators. How to explain this? Does this mean there are no need to add other indicators for yield mapping?

**Response to comment 4:**

Thank you very much for your comments and suggestions.

Only about half of the models which the importance of CEC+TI are obviously more than 50%, which suggests the importance of other indicators (CGP, EGP, CEC) still account for approximately 50%. We have attempted to estimated rice yield only by predictors of CEC+TI for some high proportioned importance and low proportioned importance cases. All the results were worse than the original models. For the high proportioned importance cases, the accuracy of only input CEC+TI predictors decreased less than that of low proportioned importance ones.

It is generally accepted that CEC+TI shows a high proportioned importance on yield estimates, which is in agreement with the findings from Huntington et al. (2020), Cao et al. (2021) and Ray et al. (2019). The three static predictor, *Lat*, *Lon*, and *Ele*, are the most basic geographical environment for rice growing and TI is used to replace the influence on rice yield of long-term agronomic technology improvements and varieties renewal because of the management data at a larger scale unavailable. Agronomic technology and varieties renewal are essential for rice yield compared with climate change and can offset the negative impacts of climate change according to the related studies (Yu et al., 2012; Ladha et al., 2021). Therefore, it is reasonable to have a high proportion for the importance of CEC+TI.

[Figure]

**Figure C1: Accuracy of AsiaRiceYield4km and only CEC+TI predictors for rice yield estimation.**

References:

Cao, J., Zhang, Z., Luo, Y., Zhang, L., Zhang, J., Li, Z., and Tao, F.: Wheat yield predictions at a county and field scale with deep learning, machine learning, and google earth engine, Eur. J. Agron., 123, 126204, https://doi.org/10.1016/j.eja.2020.126204, 2021.

Huntington, T., Cui, X., Mishra, U., and Scown, C. D.: Machine learning to predict biomass sorghum yields under future climate scenarios, Biofuels Bioprod. Biorefining, 14, 566–577, https://doi.org/10.1002/bbb.2087, 2020.

Ladha, J. K., Radanielson, A. M., Rutkoski, J. E., Buresh, R. J., Dobermann, A., Angeles, O., Pabuayon, I. L. B., Santos-Medellín, C., Fritsche-Neto, R., and Chivenge, P.: Steady agronomic and genetic interventions are essential for sustaining productivity in intensive rice cropping, Proc. Natl. Acad. Sci., 118, e2110807118, 2021.

Ray, D. K., West, P. C., Clark, M., Gerber, J. S., Prishchepov, A. V., and Chatterjee, S.: Climate change has likely already affected

global food production, PloS One, 14, e0217148, https://doi.org/10.1371/journal.pone.0217148, 2019.

Yu, Y., Huang, Y., and Zhang, W.: Changes in rice yields in China since 1980 associated with cultivar improvement, climate and crop management, Field Crops Res., 136, 65–75, 2012.

**Comment 5:** (5) When the model is applied for yield estimation during different growing season, does the pixel level cropping intensity map used or it is mainly based on the majority of rice cropping patterns in each administrative unit? The uncertainty of season rice yield might exceeded the uncertainty of the model due to the biased seasonal rice map.

**Response to comment 5:**

Thanks very much for your constructive comment.

In this study, we identified rice cropping intensity at the administrative scale due to the unavailable of suitable gridded rice cropping intensity maps. Although several large scale gridded cropping intensity maps were generated recently, they still cannot distinguish different crops especially for rice (Han et al., 2022; Liu et al., 2021). In this study, RiceAtlas, the most comprehensive and detailed spatial dataset on rice cropping intensity, was used. This dataset is nearly ten times more spatially details and has nearly seven times more spatial units compared with others (Laborte et al., 2017) which can reflect more explicit rice cropping intensity at administrative scale. At gridded scale, only pixels of rice area located in these minimum administrative unit with available seasonal rice yield were mapped. Besides, according to Response to comment 4 and Sect. 2.3.1, the pixels passed the inflection, and threshold detection were used for model training which suggest that these pixels are relative pure and can mitigate the uncertainty of rice seasons. We admit that the administrative rice crop intensity will introduce uncertainty for yield estimation, and such uncertainty of rice crop intensity had been added into Sect. 4.3 (**Lines 439-442**).

Reference:

Han, J., Zhang, Z., Luo, Y., Cao, J., Zhang, L., Zhuang, H., Cheng, F., Zhang, J., and Tao, F.: Annual paddy rice planting area and cropping intensity datasets and their dynamics in the Asian monsoon region from 2000 to 2020, Agric. Syst., 200, 103437, https://doi.org/10.1016/j.agsy.2022.103437, 2022.

Laborte, A. G., Gutierrez, M. A., Balanza, J. G., Saito, K., Zwart, S. J., Boschetti, M., Murty, M. V. R., Villano, L., Aunario, J. K., Reinke, R., Koo, J., Hijmans, R. J., and Nelson, A.: RiceAtlas, a spatial database of global rice calendars and production, Sci. Data, 4, 170074, https://doi.org/10.1038/sdata.2017.74, 2017.

Liu, X., Zheng, J., Yu, L., Hao, P., Chen, B., Xin, Q., Fu, H., and Gong, P.: Annual dynamic dataset of global cropping intensity from 2001 to 2019, Sci. Data, 8, 283, https://doi.org/10.1038/s41597-021-01065-9, 2021.

**Comment 6:** (6) Any possibility to use some in-situ collected actual yield data to validate the yield map?

**Response to comment 6:**

Thanks very much for your constructive suggestions.

Although it's difficult to collect in-situ yield for such a larger area, a fraction of in-situ single rice yield data is available from 1995 to 2015 in China. These data are obtained from China agro-meteorological stations, which are maintained by China Meteorological Administration (CMA) (http://data.cma.cn/). Fig. C2 presents the locations of the 47 agro-meteorological stations.

Fig. C3 shows that AsiaRiceYield4km was well consistent with in-situ yield as the average $R^2$ was

0.55 during 1995-2015. Moreover, the $R^2$ at the specific years could be as high as 0.72 at 2000, followed by 0.69 at 2005 and 0.68 at 2010. Besides, the *RMSE* was lower than 600 kg/ha at 2005, followed by 714 kg/ha at 2000 and 899 kg/ha at 2010. Therefore, the in-situ validation results were well satisfactory.

The *RMSE* for all years (Fig C3a) was somehow large (1019 kg/ha). Several reasons might cause such bias: the rice area planted at the agro-meteorological stations was generally lower than 0.015km$^2$, largely smaller than our pixel size (4×4km, 16 km$^2$). Besides, rice at agro-meteorological stations was well managed, thus such in-suit yields failed in characterizing those records at an administrative scale. Overall, the scale differences might be attributed as the main reason for the validation uncertainties.

[Figure]

**Figure C2: Location of the selected agro-meteorological stations.**

[Figure]

**Figure C3: (a) Accuracy between AsiaRiceYield4km and in-situ yield for all years. (b) The accuracy between AsiaRiceYield4km and in-situ collected actual yield in 2000, 2005 and 2010.**

Specific comments:

**Specific comment 1:** (1) Page 4 Line 106, what do you mean by 27 seasons?

**Response to specific comment 1:**

Thank you.

Here, 27 seasons refers to 27 different rice-cropping periods in 14 countries. However, we have to admit that "season" might confuse readers. Therefore, we've changed "season" into "case" and added the explanation for "case" (one specific rice-cropping period in a country). Relevant sentences in the manuscript were also modified.

**Specific comment 2:** (2) The authors collected many rice yield data from different sources. Please add more detailed information of the yield data including the spatial units, temporal extent, etc.

**Response to specific comment 2:**

Thanks very much for your constructive comment.

We have added detailed administrative scale and temporal information in the revised Supplement Table S1.

**Specific comment 3:** (3) Page 10, Line 229 – 234, the dataset was first divided into two parts according to the administrative units. 80% of the administrative units were randomly selected as training and validation among which 70% of samples were used for training and 30% were used as validation sets. In this case, the training samples were not 56% of the whole dataset. Same for validation and testing. Please make it more clear for readers.

**Response to specific comment 3:**

Thanks very much for your constructive comment.

The dataset division in the original manuscript is incomprehensible. For each case, 20% of the subset was selected randomly by administrative units. The rest of the 80% was split into 70% for training and 30% for validation. One sample is one administrative unit in one year with several predictors. Therefore, the training, validation, and testing dataset were 56% (80%×70%), 24% (80%×30%), and 20% (20%×100%) of the whole data, respectively (Fig. C4). The original flowchart was misleading. We have reorganized the expression of the dataset division (**Lines 241-245**) and redrawn the flowchart of the dataset division (Fig. 2 step3).

[Figure]

**Figure C4: Dataset division rules**

**Specific comment 4:** (4) Add more testing results for other years. The authors estimated rice yield for Asia for 1995-2015 but was insufficiently validated and tested for different years. Also, the temporal changes of rice yield should be added to result and discussion sections.

**Response to specific comment 4:**

Thanks very much for your constructive comment.

    We have added the temporal comparison of AsiaRiceYield4km and observed yields in Sect. 3.2 (**Lines 338 to 345**) for all years to validate the temporal accuracy of our results. Moreover, temporal variation analysis of rice yield was also included in Sect. 3.4 (**Lines 381-387**).

---

## Author Response (AR2)

**Review #2:**

**General Comment:** The authors addressed all my previous comments and I only have a few minor comments for the revised manuscript.

**Response to general comment:**

We are grateful for anonymous referee #2's recognition of this study's importance. We carefully revised our manuscript and provided a point-by-point response below. We have addressed all points raised in the revised manuscript.
* * *
Note: The individual comments (shown in black) are listed below including our responses (shown in blue). Line numbers (shown in **blue and bold font**) that we mention in this comment refer to our revised manuscript with all markup version.

**Comment 1:** Uncertainty analysis part need to be enhanced. First, it is strongly recommended to add an uncertainty map showing the degree of uncertainty. Second, some sources of the uncertainty are not mentioned. For instance, when you use the union rice area of 2000 – 2002 for the year before 2000, it also introduces some uncertainty.

**Response to comment 1:**

Thank you very much for your comments and suggestions.

We have enhanced Sect. 4.3 Uncertainty analysis. First, we added an uncertainty map based on the relative RMSE (*RRMSE*) to show the degree of the spatial uncertainty of AsiaRiceYield4km (**Line 431 - 436**). And the processing step of the uncertainty spatialization was added into Sect. 2.3.3 (**Line 295 - 300**). Second, the uncertainty of the union rice area was added into the manuscript (**Line 440 - 442**).

**Comment 2:** Some technical processing steps in the methodology is not clearly documented. For example, all the grid values were aggregated to administrative scale but it is not described that how the aggregation was done. Did you average all values over rice pixels or sum up of all rice pixel values?

**Response to comment 2:**

Thanks very much for your constructive comment.

We have improved the description of the technical process in our study. For the grid scale to administrative scale, all the grids located in one administrative unit were aggregated to this administrative unit which the values were averaged. We have described this step in detail (**Line 218 - 220**).

**Comment 3:** Line 130-131, Specifically, the union area of 2000, 2001, and 2002 was also applied to the years before 2001 because of the unavailable rice maps. Should 'before 2001' be 'before 2000'?

**Response to comment 3:**

Thank you.

In this study, the rice area union of the three years was used to represent the rice area of the current year. For 2000, the rice area is the union of 1999, 2000, and 2001 while the area of 1999 was unavailable.

The word "before" is misleading. To make it clear, we have changed "before 2001" to "from 1995 to 2000".